# Effect of Early Basal Leaf Removal on Phenolic and Volatile Composition and Sensory Properties of Aglianico Red Wines

**DOI:** 10.3390/plants11050591

**Published:** 2022-02-22

**Authors:** Debora Iorio, Giuseppe Gambacorta, Luigi Tarricone, Mar Vilanova, Vito Michele Paradiso

**Affiliations:** 1Department of Soil, Plant and Food Sciences, University of Bari Aldo Moro, Via Amendola, 165/A, 70126 Bari, Italy; d.iorio1@studenti.uniba.it; 2CREA, Council for Agricultural Research and Economics, Research Center for Viticulture and Enology, Via Casamassima 148, 70010 Bari, Italy; luigi.tarricone@crea.gov.it; 3Instituto de Ciencias de la Vid y del Vino (ICVV), Consejo Superior de Investigaciones Científicas, CSIC—Universidad de La Rioja-Gobierno de La Rioja, Carretera de Burgos Km 6, 26080 Logroño, Spain; 4Department of Biological and Environmental Sciences and Technologies, Laboratory of Agri-Food Microbiology and Food Technologies, University of Salento, S.P. 6, Lecce-Monteroni, 73100 Lecce, Italy; vito.paradiso@unisalento.it

**Keywords:** early defoliation, phenolic profile, volatile profile, sensory analysis

## Abstract

The aim of this work was to study the influence of early basal leaf removal on Aglianico wines produced in the Apulia region (Italy). Three treatments were carried out, where 100% of fruit-zone leaves on the north (DN), south (DS) and on both sides of the canopy (DNS) were removed. A control (CT), where all basal leaves were retained, was also performed. Instrumental (HPLC-DAD-MS and GC-MS) and sensory analysis (QDA) were used to evaluate the treatment effect on the phenolic and volatile compositions and on the sensory descriptors of wines. DNS reached the highest amounts of phenolic compounds, showing a change in the phenolic pattern from flavonols and anthocyanins. Moreover, leaf removal influenced the levels of 37.8% of volatile compounds, quantified by increasing the concentration when early leaf removal was applied on the north side of the canopy (DN), with respect to the south (DS) and both sides (DNS). In the sensory analysis, Aglianico wines were defined by 16 sensory attributes with GM > 30%, where the highest values were reached for defoliation treatments vs. control. In conclusion, early leaf removal treatments allowed us to modulate the phenolic and volatile concentrations of Aglianico wines.

## 1. Introduction

In the last decade, new viticulture techniques have been developed around the world for cost-effective canopy management, with the aim of improving grape and wine quality. Early basal leaf removal is an innovative viticulture practice aimed, on the one hand, at modulating the microclimate around the bunch and therefore reducing the incidence of bunch rot; on the other hand, it is aimed at enhancing the quality of grape and wine [1,2,3,4]. In previous studies, early leaf removal induced smaller and looser clusters that were less susceptible to Botrytis rot [2,4,5]. Moreover, this practice determines specific transcriptional modifications, involving the ripening program and the flavonoid metabolism [6]. Various effects of early leaf removal were reported, regarding concentrations of soluble solids, phenols and anthocyanins in grapes [1,2] and wines [5] from early defoliated vines. The expected positive impact of early leaf removal on grape and wine composition is based upon its effects on leaf/fruit ratio, canopy porosity, fruit (cluster and berry) exposure [4] and skin/berry ratio [7]. Furthermore, early leaf removal could be applied by a defoliator machine for cost-effective yield control with improved grape [1] and wine composition [5] and aroma attributes [8].

The modulation of phenolic contents and profiles is a main aim of viticultural practices in light of their cascading effect on the quality of grapes and finally of wines [8,9]. Several attempts have been made to adopt suitable practices with this aim [10,11]. Such viticultural practices are being also considered as tools for the adaptation to climate changes, that are severely impacting the phenolic ripening and equilibrium in grapes [12,13,14,15]. Improvement of wine bouquet is also of great interest to viticulturists and winemakers due to their importance to wine quality. Generally, wine aroma can be categorized as varietal aromas (terpenes, norisoprenoids and methoxypyrazines), fermentation aromas (higher alcohols and their acetates, as well as fatty acids and their ethyl esters) and aging aromas (volatile phenols). Varietal aromas of wine mainly derive from grapes and are subjected to genotypic and environmental factors (light, temperature and water availability) [16]. Given this, basal leaf removal can be an effective practice for directly modifying wine varietals aromas. Fermentation aromas are formed via fatty acid metabolism or amino acid metabolism by yeast activity during fermentation [17]. Furthermore, several studies have demonstrated that fatty acids and amino acids are sensitive to environmental factors [18,19,20]. Fatty acids in berries have shown diverse behaviors in different training systems [18], and concentrations of amino acids in berries have been altered by sunlight exposure [19,20]. Thus, it is possible that fermentation aromas can be affected by basal leaf removal by altering their substrate levels. Many studies have recently been conducted to investigate the influence of basal leaf removal on volatile compounds in grape and wine, but discrepancies exist among these studies. The controversial results across these studies indicate that the grape cultivar or clone [21,22], the climate condition [23], grape maturity [24] and the timing and severity of defoliation [23,25,26] might be responsible for the varied effects of basal leaf removal on the aromatic properties of grape and wine.

Instrumental and sensory analyses allow researchers to study the phenolic and aromatic composition of the wines. Volatile and non-volatile components of the wine can be identified and quantified by chromatographic techniques, and the sensory impact of volatiles depends on their perception thresholds [27]. On the other hand, sensory analysis allows detection and description of qualitative and quantitative sensory components of a product by a trained panel of judges [28]. Sensory descriptive analysis [29] is one of the most comprehensive and informative tools used in sensory analysis. This technique can provide complete sensory description of a product such as wine. Information from instrumental and sensory data, is very important to establish the composition of wine. The relationship between instrumental and sensory data has been extensively studied [16,26,27,30,31,32].

In this sense, the aim of this work was to study the influence of early leaf removal in the vineyard on Aglianico wines quality from Apulia region (Italy). Instrumental (HPLC-DAD-MS and GC-MS) and sensory analyses (QDA) were performed to evaluate this effect on phenolic and volatile composition and sensory descriptors of wines.

## 2. Results

### 2.1. Oenological Parameters

Table 1 shows the influence of the leaf removal treatments on the oenological parameters of Aglianico wines. ANOVA results, by treatment, are also shown. In general, early leaf removal led to wines of higher alcohol, and more total polyphenol index, whereas pH and titratable acidity and malic, tartaric, citric and acetic acids remained generally unaffected. Similar results were found in Tempranillo wines from Spain, where higher alcohol content, more intense colors and a larger total polyphenol index were shown when pre-bloom leaf removal was applied [26]. Our results also coincide with Diago et al. [8], who reported that mechanical leaf removal was more effective in reducing yield, cluster weight and number of berries than manual leaf pulling, by affecting the fruit microclimate. On the other hand, no increase in alcohol contents was observed after early leaf removal in Gamay, Nero d’Avola, Graciano and Carignan wines [5,24,33]. A recent meta-analysis [3] summarized the outputs of research about the effects of early leaf removal on grape production and quality parameters. The most relevant findings showed the lowering of bunch rot disease (−61%) and the increase in berry total soluble solids (+5.2%, °Brix), which was related to the increase in the leaf-to-fruit ratio. Regarding the response of other quality indices, rootstock and variety were the most relevant variables influencing the effects of early leaf removal, while the role of climate was less relevant.

### 2.2. Analysis of Polyphenols

Table 2 shows the influence of the defoliation treatments on the polyphenolic indices of the wines. All the indices related to polyphenols showed a significant increase in wines from early defoliated vines. Some variations of the effect of defoliation were observed based on the canopy side involved. When early leaf removal involved the north side, the highest levels of anthocyanins, total flavonoids and flavans reactive with vanillin (FRV) were reached. On the other hand, wines from vines defoliated on the south side presented the highest levels of proanthocyanidins. Leaf removal on both sides led to the highest levels of all the classes, except FRV, and therefore led to the maximum content of total polyphenols in wines. The FRV/P ratio, representing the degree of tannin condensation, was found to be lower in defoliated wines, especially on the south and north–south sides (0.43–0.46 versus 0.56 for CT), which implied a reduction in tannin reactivity and a predisposition to color and tannin stabilization of wines [34].

Individual polyphenols were identified by HPLC-MS (Table 3) and quantified by HPLC-DAD (Table 4). Figure 1 reports the biplot of the first two principal components obtained from the PCA of phenolic profiles. Leaf removal on both sides determined a general increase in wines of compounds from flavonoid biosynthesis compared with control wines, as a consequence of light exposure [35].

Selective defoliation on either side impacted on wine phenolic profile as a consequence of a possible metabolic shift of phenolic compounds biosynthesis, from flavonol biosynthesis to anthocyanin biosynthesis, especially when defoliation on the north side was adopted. This could be attributed to the increase in anthocyanin biosynthesis with a competitive effect towards flavonols pathway [36]. DS wines only presented slight changes compared with control, while DN wines showed more evident differences. Compared with the effects of partial leaf removal on anthocyanin profiles of grape berries [37], some discrepancies were observed. In fact, leaf removal on the south side of the canopy determined higher levels of free monoglycosides in Aglianico berries [37] with respect to leaf removal on the north side. In wines, the opposite trend was observed.

On the other hand, higher levels of pyranoanthocyanins (vitisin A and vitisin B) were observed in DS wines compared with DN wines. Therefore, leaf removal treatments also affected oxidation-mediated pigment stabilization, besides pigment profiles. An effect of leaf removal treatments on vitisins was previously reported by other authors [38]. The meta-analysis conducted by VanderWeide [3] reported that early leaf removal altered secondary metabolites (e.g., anthocyanins and total polyphenols) to a greater extent than soluble solids, the mean increase observed was not statistically significant, mainly due to cultivar variability, as well as the differences in the analytical protocols adopted in the different studies. Kemp et al. [39] reported increased proanthocyanin levels derived from early leaf removal in Pinot noir. These results were confirmed for Pinot noir by Verdenal and colleagues, who found an increase on total polyphenols and total anthocyanins [40]. Verdenal et al. [33] carried out another extensive survey on the effect of early leaf removal on five different cultivars, both red and white, and observed relevant differences in the responses among cultivars. However, due to preflowering leaf removal, the red wines were often preferred for their color. Wines from Dalmatian cv. Trnjak had higher anthocyanin concentration when obtained upon early leaf removal [41]. Increase in anthocyanin and total phenols was also observed in Carignan and Graciano wines as an effect of early defoliation [5]. Basal leaf removal increased total amount of anthocyanins, flavonoids, polyphenols and color intensity in wines from the Sicilian cultivar Nero d’Avola; although, the effect strongly depended on the time of grape harvesting [24]. Torres et al. [9] recently reported that early defoliation combined with shoot thinning led to Cabernet Sauvignon wines with higher total polyphenols; shifts in the profiles of anthocyanins, flavonols, flavan-3-ols and proanthocyanidins were also observed. However, Guidoni et al. [4] reported an improvement in Barbera skin anthocyanin and polyphenol composition only in the coolest and rainiest year—the least suitable weather for Barbera ripening. Cluster direct exposure to sunlight increases cluster temperature, frequently causing berry withering or sunburn, and modifying the accumulation of some berry quality components. Therefore, in microclimatic conditions, unfavorable for exposed vineyards, leaf removal may improve grape health and quality [4].

### 2.3. Wine Volatile Composition

Figure 2 shows the effect of early leaf removal (DN, DS and DNS) on the concentration of the volatile composition of Aglianico wine. They have been grouped into several groups: alcohols, C6 compounds, terpenes + C13 norisoprenoids, esters + acetates, volatile acids, aldehydes, volatile phenols, and lactones. Early leaf removal induced significant changes in the concentration of four groups of compounds (C6 compounds, terpenes + C13 norisoprenoids, volatile acids, and volatile phenols) increasing their concentration with respect to the control. In the same way, a trend to increase the concentration on the other groups of compounds studied was observed when early leaf removal was applied in Aglianico. Vitis vinifera L. cv. Nero d’Avola, submitted to early defoliation treatment, showed significantly higher amounts of most volatile constituents, such as acids, furfural aldehydes and C13-norisoprenoids, than did the control ones [24].

Additionally, slight variations of volatiles by the effect of early defoliation were observed based on the canopy side involved. Thus, Aglianico wines from early leaf removal in the north side (DN) showed a tendency to increase the concentration of all volatile families of compounds with exception of lactones, which was higher when defoliation was made in both the north and the south side of the canopy (DNS). According to these results, early leaf removal induced the increase in the concentration of all families of volatile compounds, quantified in Tempranillo wine from Southern Spain, with exception of lactones [42]. Other researchers showed that leaf removal induced an increase in grape and wine volatile composition [26,43].

Table 5 presents the influence of the early leaf removal treatment (DS, DN and DNS) and control on the individual volatile compounds identified and quantified by GC-MS in Aglianico wines (expressed as μg/L); one-way ANOVA results, by treatment, are also shown. A total of 37 volatile compounds were identified and quantified by GC-MS in Aglianico wines. Alcohols were the largest group of volatile compounds accounting for all leaf removal and control treatments with 9 compounds quantified (>78% of the total volatile concentration), followed by ethyl esters + acetates, represented by 9 compounds (>13% of total volatile concentration).

Results of the ANOVA showed the effect of treatments on 37.8% (14 out 37 compounds) of the volatiles identified and quantified. Leaf removal treatments led to wines with the significant highest concentrations of 13 volatile compounds vs. control, mainly ethyl esters and acetates. Thus, early leaf removal induced the increase of three alcohols (3-methyl-1-pentanol, 2,3-butanediol and 1-octanol) one C6 compound (1-hexanol), one terpene (E-8-hydroxy linalool), four esters + acetates (hexyl acetate, diethyl succinate, diethyl malate and ethyl myristate), three volatile acids (octanoic, nonanoic and decanoic acids) and one volatile phenol (4-vinylphenol). From the phenol volatiles group, 4-ethylphenol was not identified when early leaf removal was applied. The increase in acetates concentration when early defoliation was applied was also observed in Tempranillo wines from Northern Spain [26]. In the same work, it was observed that early leaf removal induced a significant reduction in C6 compounds. However, Tempranillo wine from Southern Spain showed higher concentration of C6 compounds when pre-bloom basal leaf removal was applied vs. control [42]. In this study, according with our results, the highest concentration was observed for 1-hexanol. In Istrian Malvasia wine, the effect of early defoliation was higher than defoliation at véraison, where wines from pre-bloom defoliation increased ethyl esters and higher alcohols concentration [44].

On the other hand, the effects of defoliation depended on amount and type of leaves removed and on defoliation timing [45]. The grapevine canopy consists of leaves of different ages, which are subjected to variable light intensities during the entire growth season, affected by photosynthesis, transpiration and microclimate [46]. In fact, the high light intensity and temperature induced by excessive defoliations may reduce the skin color in red cultivars [47]; while several authors have observed a positive effect of light penetration on grape quality [24,26,44), the negative influence on vine metabolism could be due to the effect of high temperatures in some semiarid regions affecting to wine aroma and color [42].

The effect of early defoliation based on the canopy side involved was observed on individual compounds. Thus, Aglianico wines from early leaf removal on the north side (ND) showed a trend to increase the mayor number of compounds quantified, with exceptions of 2-methyl-1-propanol, 3-methyl-1-propanol, 3-oxo-7,8-dihydro-a-ionol, isoamyl acetate and hexyl acetate, which were higher when early leaf removal was made on the south side (SD) of the canopy. Moreover, three compounds (2,3-butanediol, 4-ethylguaiacol and butyrolactone) showed a trend to increase their concentration when early defoliation was performed on both sides of the canopy (DNS). In this sense, it is known that the sunlight exposure affects the cluster temperature, influencing the degradation of malic acid, increasing the sugar/acid ratio and significantly affecting the levels of varietal aroma compounds in grape berries [48].

### 2.4. Sensory Analysis

The effect of early leaf removal on Aglianico wines was evaluated by sensory descriptive analysis. The sensory profile of Aglianico wines was characterized by 27 descriptors belonging to odor (17 descriptors), taste (9 descriptors) and total value. The frequency (F), intensity (I) and geometric mean (GM) of the different descriptors are shown in Table 6.

From odor descriptors, the most relative intensities were showed by odor intensity (>63%), odor quality (>54%), fruity (>45%) and red fruits (>46%), where wines from DNS treatments showed the highest values (70, 71, 54 and 71, respectively). With respect to taste descriptors, DNS wines showed the highest relative intensity values for quality (63%), body (60%) and persistence (64%). The wines from early defoliation on the north side (DN) were the most astringent. 

With respect to relative frequency, wines from early leaf removal, mainly DNS, exhibited the highest values (>55%) of eight odor descriptors (intensity, quality, fruity, floral, herbaceous, red fruit, spicy and phenolic). An increase in fruity and floral odor was observed in Tempranillo wine when early defoliation was applied [42]. In taste, all descriptors showed relative frequency > 50% with exception to balanced descriptor. In general, the most frequency of these descriptors were found for wines from DNS treatment, with the exceptions of salt (DN and DS) and bitter (CT and DN). Pinot noir wines from the preflowering defoliation treatment were described as less fruity and more herbaceous in comparison with those of the other treatments. In terms of mouth feel, these tended to have more volume and intensity [33].

In respect to relative intensity of global quality, wines from the DNS treatment showed the highest value (67.5%) and all wines exhibited 100% relative frequency.

The intensity (I) and frequency (F) of each attribute permitted the geometric mean (GM) to be obtained. GM (%) was calculated for each descriptor as a square root of the product between the relative intensity and relative frequency. In this study, descriptors with GM > 30% were considered the highest contributors. Eight sensory attributes with GM > 30% defined the aroma of Aglianico wines. In taste, eight attributes showed GM > 30%. Odor and taste profiles in relation to GM (%) are showed in Figure 3. The highest % GM of total value was also reached by wines from DNS treatment. Defoliation enhances grape quality [49,50] by improving berry color and flavor [51], total anthocyanin and phenolic concentrations, color intensity and sensory quality of wines [52]. The wine tasting confirmed the enhancement of wine aromas and taste through significant changes in the concentration of volatile compounds, according to results observed by other researchers [23,30]. These results explained that in spite of the effects of defoliation on wine volatiles, results of sensory analysis depend on perception of the interactions among volatile compounds and on the threshold values of each compound [53].

## 3. Materials and Methods

### 3.1. Vineyard, Leaf Removal Treatment and Vinification

Early basal leaf removal experiments were conducted in commercial Aglianico (*Vitis vinifera* L.) vineyards situated in Apulia region in Southern Italy (CGDO Castel del Monte area) during the 2018 season (Corato, lat: 41°04′35″ N; long. 16°21′46″ E 354 m a.s.l.). Approximately 15 days before flowering (middle of May BBCH 57), the following 4 early leaf removal treatments (leaf removal of the basal part of the shoot up to the last cluster) were manually applied: CT—no leaf removal or non-defoliated vines, where all basal leaves were retained in each shoot; DS—100% of fruit-zone leaves on each shoot were removed from the south canopy side; DN—100% of fruit-zone leaves on each shoot were removed from the north canopy side; DNS—100% removal of fruit-zone leaves on both the north and the south sides of the canopy.

Six adjacent rows were selected to set up a randomized complete block design, with two rows as a block. Within each 2 rows, 3 sections of 18 vines per plot were tagged and randomly assigned to the leaf removal treatments, imposed with 54 vines for each treatment.

Per trial, about 80 kg of grapes at commercial maturity were immediately submitted to winemaking at the experimental winery of Bari University according to Gambacorta et al. [54] protocol, with some modifications. In brief, grapes of each trial were de-stemmed, crushed and transferred in 100 L vertical stainless steel vats, and potassium metabisulphite (6 g/100 kg), yeast (*Saccharomyces cerevisiae* var. Bayanus, Mycoferm CRU05, 20 g/100 kg) and yeast activator (preparation based on ammonium sulphate, diammonium phosphate, chemically inert filter and as dispersing agent, Vitamin B1, Enovit, AEB, Italy) were added. Nine days of maceration were applied with two punch-downs per day. When maceration was concluded, free-run wine was recovered, and pomace was gently pressed for obtaining press-run wine using 80 L stainless steel hydro press. The two wine fractions were blended after 2 weeks of racking was performed to eliminate gross lees. Wine was bottled after 6 months, without any treatment. Three replications of wines were analyzed.

### 3.2. Enological Parameters

Sugar concentration, pH, ethanol, titratable acidity, citric, tartaric, malic and acetic acids and total polyphenol index (TPI) in wines were analyzed in triplicate by using a Foss WineScan FT 120, as described by the manufacturer (Foss, Hillerød, Denmark).

### 3.3. Analysis of Polyphenols

The phenolic composition of wines was analyzed by spectrophotometry as described by Di Stefano and Cravero [55]. Color intensity (CI), given by the sum of absorbances at 420, 520 and 620 nm, and hue (ratio of absorbances 420 nm/520 nm) were analyzed by the Glories method [56].

Antioxidant activity (AA) was assessed using ABTS [2,2′-azino-bis(3-ethylbenzothiazoline-6-sulfonic acid)] assay [57], and results were expressed as μM Trolox equivalent antioxidant.

The polyphenolic profile of wines was determined by LC-MS/MS. A Dionex Ultimate 3000 LC System (Thermo Fisher Scientific, MA, USA) comprising a quaternary pump, an autosampler, a column oven and a DAD detector. The LC system was interfaced with a LTQ Velos Pro Linear Ion Trap mass spectrometer (Thermo Fisher Scientific, MA, USA) through a HESI interface. The samples, previously filtered on 0.22 μm regenerated cellulose membrane, were injected into a C18 Hypersil GOLD aQ column (100 mm × 2.1 mm × 1.9 µm, Thermo Fisher Scientific, MA, USA). The mobile phase was constituted by water acidified with 10% formic acid (A) and acetonitrile with 0.1% of formic acid (B). The flow rate of the mobile phase was 0.3 mL min^−1^, and the injection volume was 5 μL. A gradient-elution program was as follows: linear gradient from 2% B to 70% B, 0–20 min, isocratic of 70% B, 20–24 min, linear gradient from 70% B to 2% B, 24–34 min. The mass spectrometer conditions were as follows: spray voltage +2.5 kV, sheath gas 30 psi, auxiliary gas flow 15 arbitrary units, capillary temperature 320 °C, capillary voltage + 95 V, tube lens + 170 V, skimmer + 38 V, and heater temperature 280 °C. Samples were analyzed in MS/MS in data dependent mode, with a full scan in the range 150–1200 *m*/*z*, mass spectrometry data were acquired in both positive and negative ion mode. Malvidin-3-O-glucoside, gallic acid, caftaric acid, quercetin-3-glucuronide, miricetin, quercetin and catechin were identified by comparing elution times, molecular ions, and MS/MS fragmentation patterns of the experimental spectra with those obtained by pure standards, whereas other compounds were tentatively identified by data reported from the literature [58,59,60,61]. Quantitative analysis was carried out using a diode array detector at wavelengths 280 nm (flavanols), 330 nm (phenolic acids), 350 nm (flavonols) and anthocyanins (520 nm). Calibration curves of malvidin-3-glucoside (for anthocyanins), gallic acid (for phenolic acids) and quercetin (for flavanols and flavonols) were built (R2 = 0.9975, R2 = 0.9988, R2 = 0.9974 respectively). The analyses were carried out in triplicate and results were expressed in mg L^−1^.

### 3.4. Wine Volatile Composition

In a 10 mL culture tube, 8 mL of wine, 3 μg of internal standard (4-nonanol) and a magnetic stir bar (22.2 mm × 4.8 mm) were added. Volatiles were extracted by stirring the sample with 400 mL of dichloromethane, as described by Coelho et al. [62]. After cooling at 0 °C for 10 min, the magnetic stir bar was removed, the organic phase was detached by centrifugation (5118 *g*, 5 min, 4 °C), and the extract was recovered into a vial, using a Pasteur pipette. The aromatic extract (200 μg L^−1^) was dried with anhydrous sodium sulphate and placed in a new vial. Volatile compounds were extracted from each of the wines in triplicate.

Gas chromatographic analysis of volatile compounds was performed using an Agilent GC 6890 N chromatograph (Agilent Technologies, Palo Alto, CA, USA) coupled to mass spectrometer Agilent 5975C. A 1 μL injection was made into a capillary column, coated with CP-Wax 52 CB (50 m × 0.25 mm i.d., 0.2 μm film thickness, Chrompack). The temperature of the injector was programmed from 20 °C to 250 °C, at 180 °C min^−1^. The oven temperature was held at 40 °C for 5 min, then it was programmed to rise from 40 °C to 250 °C, at 3 °C min^−1^, then it was held for 20 min at 250 °C, and finally it was programmed to go from 250 °C to 255 °C at 1 °C min^−1^. The carrier gas was helium N60 (Air Liquide) at 103 kPa, which corresponds to a linear speed of 180 cm s^−1^ at 150 °C. The detector was set to electronic impact mode (70 eV), with an acquisition range from 29 to 360 m/z, and an acquisition rate of 610 ms.

The compounds were identified using WSearch Free Software, by comparing mass spectra and retention indices with those of pure standard compounds. Pure standard compounds were purchased from Sigma-Aldrich (Darmstadt, Germany) with purity higher than 98%. Semi-quantitative data were obtained by calculating the relative peak area in relation with internal standard (4-nonanol).

### 3.5. Sensory Analysis

The sensory analysis was performed by 10 panelists from Rias Baixas AOC (Galicia, Spain) sensory panel, 4 males and 6 females, with ages between 35 and 60 years old. All the judges were experienced tasters and all of them have previously participated in similar studies. In accordance with ISO Norm 8589, the sensory analysis was performed in a professional-standard room. The evaluation was carried out using the QDA methodology [63] to establish descriptors of the wines. The terms, balance, odor quality, taste quality and total quality, normally used by the Rias Baixas AOC panel, were added to the sensory analysis. Odor quality and taste quality were defined by the panel as absence of defects; total quality was defined as the global perception of the wines (odor and taste); balance was defined as harmony, the integration of acidity, sugar, alcohol and bitter. A constant sample volume of 30 mL of each wine was evaluated in wine taster glasses at 12 °C. During the analysis, the judges smelled and tasted the samples, and the perceived descriptors were indicated. Then, they scored the intensity of each attribute using a 10-point scale, where 10 indicated a very high intensity. The relative frequency (F), relative intensity (I) and geometric mean (GM) of the different descriptors were calculated for each wine. GM was calculated as the square root of the product between I and F, i.e., GM (%) = √(I × F) ×100, where I corresponds to the sum of the intensities given by the panel for a descriptor, divided by the maximum possible intensity for this descriptor; F is the number of times that the descriptor was mentioned, divided by the maximum number of times that it could be mentioned.

The descriptors were classified for each wine by using the GM according to the International Organization for Standardization—ISO Norm 11,035—which made it possible to eliminate the descriptors whose geometric means were relatively low. This method allowed us to consider descriptors which were rarely mentioned but which were very important in terms of the perceived intensity, and descriptors with a low perceived intensity but which are mentioned often [64].

### 3.6. Statistical Analysis

All data were analyzed using the XLSTAT-Pro 2017 statistical package (Addinsoft, Paris, France). A one-way ANOVA was used to evaluate the differences among treatments. The multiple comparison among treatments were calculated according to the least significant difference from Tukey’s test. Principal components analysis (PCA) was performed using Origin Pro 2021 (OriginLab, Northampton, MA, USA).

## 4. Conclusions

Early defoliation is a viticulture practice aimed at regulating yield components and improving grape quality. The effect of vines defoliation on Aglianico wine quality was studied. Early basal leaf removal allowed us to modulate the volatile and phenolic compounds and, therefore, the sensory properties of wines. Defoliation led to wines with higher phenolics and volatile compounds concentration. As a consequence, the sensory attribute intensities were influenced too. Selective leaf removal on one side of the canopy (either north or south) allowed us to obtain wines with different chemical and sensory properties. Aglianico wines from early leaf removal on the north side (DN) showed a trend of an increase in the major number of volatile compounds quantified. Leaf removal on both sides (DNS) led to the highest levels of total polyphenols in the wines. Early leaf removal is, therefore, effective in modulating the properties of grapes and wines.

## Figures and Tables

**Figure 1 plants-11-00591-f001:**
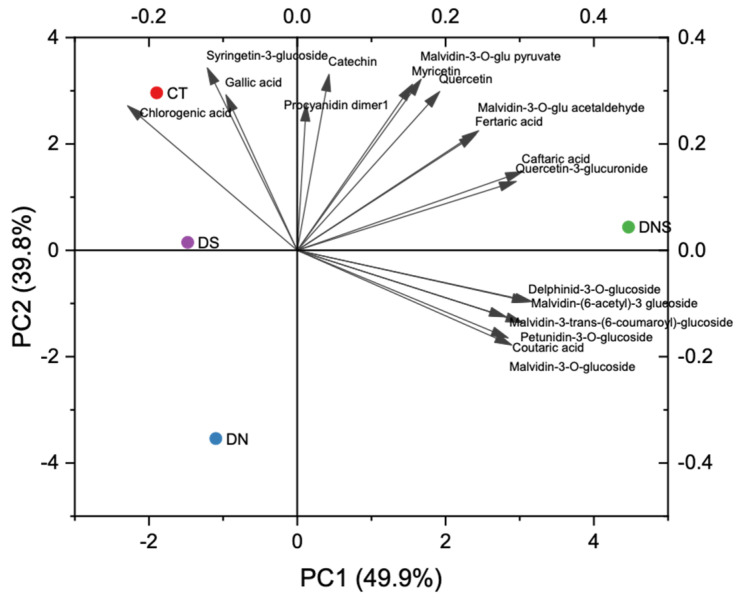
Principal components analysis of the phenolic profiles of Aglianico red wines obtained by early basal leaf removal treatments. CT—control; DN—defoliation on the north side of the canopy; DS—defoliation on the south side of the canopy; DNS—defoliation on both sides of the canopy.

**Figure 2 plants-11-00591-f002:**
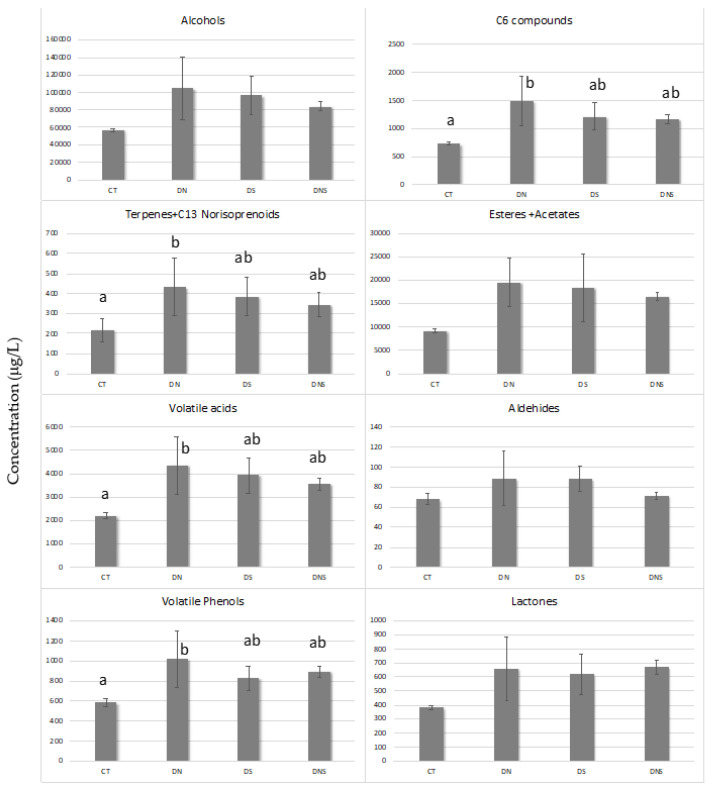
Total volatile concentration (mg/L) by families of Aglianico red wines obtained by early basal leaf removal treatments. Differences letters indicate significant differences for Tukey’s test at *p* < 0.05. CT—control; DN—defoliation on the north side of the canopy; DS—defoliation on the south side of the canopy; DNS—defoliation on both sides of the canopy.

**Figure 3 plants-11-00591-f003:**
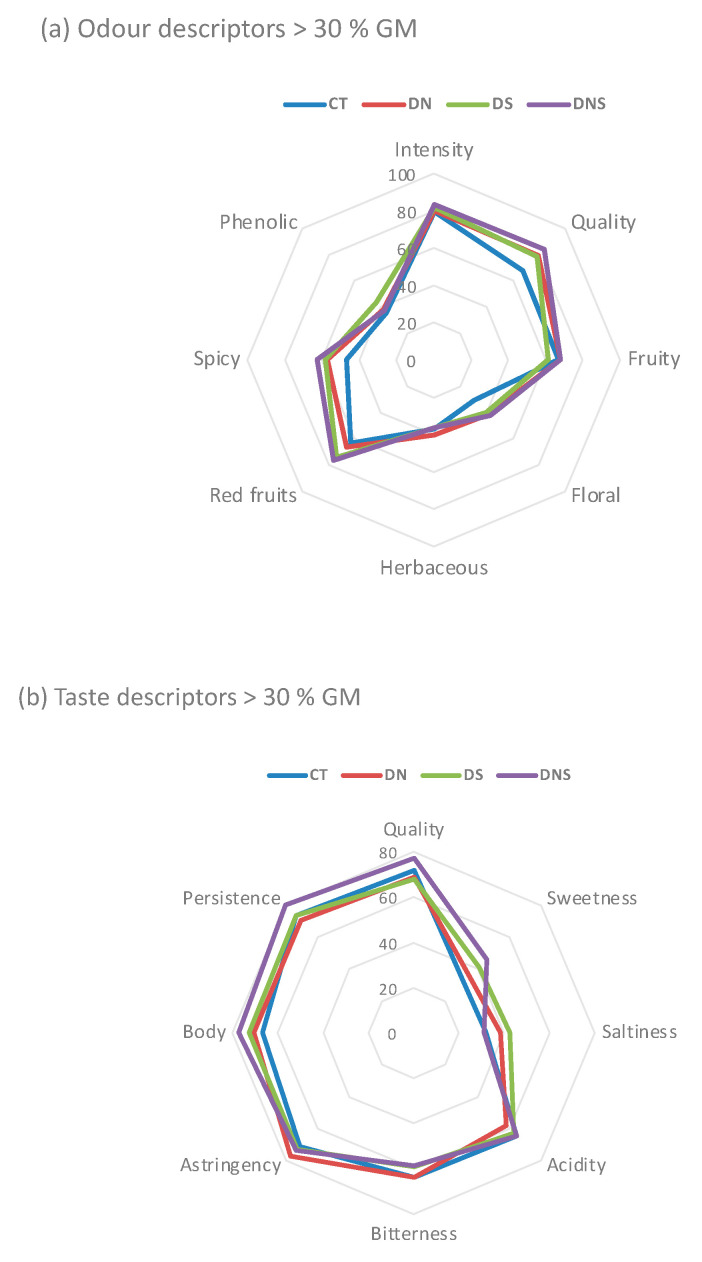
Odor (**a**) and taste (**b**) profiles (% GM) of Aglianico red wines obtained by early basal leaf removal treatments. CT—control; DN—defoliation on the north side of the canopy; DS—defoliation on the south side of the canopy; DNS—defoliation on both sides of the canopy.

**Table 1 plants-11-00591-t001:** Enological parameters of Aglianico red wines from 2018 vintage, obtained by early basal leaf removal treatments.

Parameters	CT		DN		DS		DNS		Sig.
	Mean	SD	Mean	SD	Mean	SD	Mean	SD	
Glucose + Fructose (g/L)	0.12 a	0.02	0.31 c	0.01	0.24 b	0.01	0.39 d	0.02	***
Ethanol (%vol)	13.17 a	0.03	13.52 c	0.03	13.32 b	0.06	13.82 d	0.03	***
PH	3.20	0.04	3.16	0.05	3.23	0.04	3.18	0.04	ns
Total acidity (g/L)	5.89	0.20	6.12	0.20	5.81	0.22	6.10	0.19	ns
Tartaric acid (g/L)	1.29	0.34	1.30	0.36	1.33	0.19	1.33	0.34	ns
Citric acid (g/L)	0.26	0.01	0.28	0.02	0.26	0.01	0.27	0.00	ns
Malic acid (g/L)	0.98	0.03	0.94	0.04	0.94	0.01	0.91	0.06	ns
Acetic acid (g/L)	0.09	0.01	0.07	0.01	0.07	0.01	0.08	0.01	ns
IPT	35.07 a	1.81	40.47 b	0.76	36.00 a	1.31	40.20 b	0.75	**

Different letters indicate significant differences for Tukey’s test at *p* < 0.05. Signs: **—significance at *p* < 0.01; ***—significance at *p* < 0.001; ns—not significant; CT—control; DN—defoliation on the north side of the canopy; DS—defoliation on the south side of the canopy; DNS—defoliation on both sides of the canopy.

**Table 2 plants-11-00591-t002:** Polyphenols and color indices of Aglianico red wines from 2018 vintage, obtained by early basal leaf removal treatments.

Polyphenols Indices	CT		DN		DS		DNS		Sig.
	Mean	SD	Mean	SD	Mean	SD	Mean	SD	
Total anthocyanins (mg/L)	294 c	1	447 a	4	337 b	4	449 a	4	*
Total flavonoids (mg/L)	1474 c	4	1715 a	6	1562 b	16	1735 a	9	*
Vanillin reactive flavans (mg/L)	948 b	44	1113 a	24	1037 ab	57	1014 ab	34	*
Proanthocyanidins (mg/L)	1697 c	125	2061 b	94	2241 ab	100	2369 a	121	*
Total polyphenols (mg/L)	1709 c	44	1969 b	35	1895 b	31	2159 a	40	*
Antioxidant activity (mmol/L)	10.0 b	0.5	12.2 a	0.4	12.3 a	0.4	13.3 a	0.3	*
Color intensity	1.42 c	0.01	1.61 b	0.01	1.63 b	0.03	2.01 a	0.04	*
Hue	0.53	0.01	0.52	0.01	0.53	0.01	0.52	0.01	ns

Different letters indicate significant differences for Tukey’s test at *p* < 0.05. Signs: *—significance at *p* < 0.05; ns—not significant; CT—control; DN—defoliation on the north side of the canopy; DS—defoliation on the south side of the canopy; DNS—defoliation on both sides of the canopy.

**Table 3 plants-11-00591-t003:** Identification of polyphenols of Aglianico red wines from 2018 vintage, obtained by early basal leaf removal treatments.

Compounds	RT	λ Monitored	Molecular Ion	Fragments	Standard
	min	nm	*m*/*z*	*m*/*z*	
Anthocyanins					
Delphinidin-3-O-glucoside	5.68	520	465	303	No
Petunidin-3-O-glucoside	6.57	520	479	317	No
Malvidin-3-O-glucoside	7.34	520	493	331	Yes
Malvidin-3-O-glu pyruvate	7.75	520	561	399	No
Malvidin-3-O-glu acetaldehyde	8.01	520	517	355	No
Malvidin (6-acetyl)-glucoside	9.14	520	535	331	No
Malvidin-3-trans(6-coumaroyl)-glucoside	10.38	520	639	331	No
Phenolic acids					
Chlorogenic acid	1.00	330	191	111, 173	Yes
Gallic acid	1.16	330	169	125	Yes
Caftaric acid	2.11	330	311	149, 179	Yes
Coutaric acid	3.35	330	295	163, 149	Yes
Fertaric acid	4.81	330	325	193	No
Flavonols					
Quercetin-3-glucuronide	7.65	350	477	301	Yes
Miricetin	8.59	350	317	151, 179	Yes
Syringetin-3-glucoside	8.93	350	507	345	No
Quercetin	10.35	350	301	151, 178	Yes
Flavanols					
Procyanidins dimer-1	2.29	280	577	425, 407	No
Catechin	2.92	280	289	245, 205	Yes

RT—retention time.

**Table 4 plants-11-00591-t004:** Concentration (mg/L) of wine polyphenols from Aglianico red wines obtained by early basal leaf removal treatments.

Compounds	CT		DN		DS		DNS		Sig.
	Mean	SD	Mean	SD	Mean	SD	Mean	SD	
**Anthocyanins**									
Delphinid-3-O-glucoside	4.21 c	0.09	9.44 b	0.90	3.51 c	2.13	17.08 a	2.63	*
Petunidin-3-O-glucoside	7.76 c	0.86	17.59 b	1.14	8.54 c	0.78	26.96 a	3.72	*
Malvidin-3-O-glucoside	105.16 d	2.15	207.65 b	4.77	128.38 c	8.98	269.19 a	4.77	*
Malvidin-3-O-glu pyruvate	56.72 a	3.22	27.17 c	2.41	41.42 b	3.17	59.51 a	6.64	*
Malvidin-3-O-glu acetaldehyde	39.79 b	3.90	22.81 c	1.76	42.16 b	0.87	57.18 a	1.54	*
Malvidin-(6-acetyl)-3 glucoside	9.05 c	5.32	15.14 b	2.80	9.08 c	1.69	24.28 a	2.08	*
Malvidin-3-trans-(6-coumaroyl)-glucoside	8.25 c	1.30	16.19 b	2.41	4.55 d	1.21	23.01 a	2.15	*
Total (%)	230.94 c (72)		315.99 b (82)		237.64 c (74)		477.21 a (85)		*
**Phenolic acids**									
Chlorogenic acid	1.59	0.82	0.84	0.66	1.24	0.44	0.76	0.06	ns
Gallic acid	1.04	0.36	0.42	0.53	0.42	0.12	0.51	0.2	ns
Caftaric acid	40.12	3.9	39.13	3.06	40.08	1.35	42.47	1.27	ns
Coutaric acid	9.81	2.64	10.70	0.15	10.60	1.28	11.42	2.14	ns
Fertaric acid	1.94	2.37	1.27	0.24	1.25	0.54	2.30	0.1	ns
Total (%)	54.5 (17)		52.36 (14)		53.59 (17)		57.46 (10)		ns
**Flavonols**									
Quercetin-3-glucuronide	5.25 b	0.82	4.13 c	0.8	6.15 b	0.72	8.91 a	0.27	*
Myricetin	3.09 a	0.27	1.05 c	0.13	2.90 b	0.14	3.56 a	0.34	*
Syringetin-3-glucoside	7.58 a	1.41	1.84 d	0.73	4.26 b	0.63	3.29 c	0.16	*
Quercetin	2.33 b	0.23	0.67 d	0.09	1.97 c	0.1	2.90 a	0.14	*
Total (%)	18.25 a (6)		7.69 c (2)		15.28 b (5)		18.66 a (3)		*
**Flavanols**									
Procyanidin dimer1	5.99	1.11	4.09	1.06	3.73	0.55	4.85	0.98	ns
Catechin	8.79 a	1.75	4.45 b	1.71	8.95 a	1.6	8.33 a	2.09	*
Total (%)	14.78 a (5)		8.54 b (2)		12.68 a (4)		13.18 a (2)		*
**Total Polyphenols**									
	318.47 c		384.58 b		319.19 c		566.51 a		*

Different letters indicate significant differences for Tukey’s test at *p* < 0.05. Signs: *—significance at *p* < 0.05; ns—not significant; CT—control; DN—defoliation on the north side of the canopy; DS—defoliation on the south side of the canopy; DNS—defoliation on both sides of the canopy.

**Table 5 plants-11-00591-t005:** Concentration (μg/L) of wine volatiles from Aglianico red wines obtained by early basal leaf removal treatments.

Compounds	CT		DN		DS		DNS		Sig.
	Mean	SD	Mean	SD	Mean	SD	Mean	SD	
**Alcohols**									
2-methyl-1-propanol	998.2	15.5	1744.4	650.6	1851.1	493.5	1810.7	73.5	ns
2+3-methyl-1-butanol	25,880.4	696.8	49,220.0	18,132.0	45,350.7	11,548.9	38,843.5	2471.5	ns
3-methyl-1-pentanol	32.1 a	1.4	54.8 ab	16.5	62.7 b	8.1	60.4 b	2.6	*
2,3-butanediol	108.8 a	4.0	273.8 b	100.3	301.7 b	53.2	360.1 b	48.4	**
1-octanol	158.5 a	9.8	334.7 b	91.1	302.8 ab	64.9	290.1 ab	20.3	*
3-methylthiopropanol	222.3	15.8	429.5	163.9	353.0	114.3	300.9	21.9	ns
Benzyl alcohol	24.3	1.2	46.1	17.1	43.7	11.3	38.5	2.7	ns
2-phenylethanol	29,468.0	912.4	52,814.4	16,824.5	48,537.4	10,200.5	42,152.7	2555.7	ns
**C6 compounds**									
1-hexanol	669.0 a	30.1	1366.4 b	390.2	1109.2 ab	212.9	1071.1 ab	68.1	*
E-3-hexenol	25.2	2.0	46.3	17.1	42.2	18.4	33.9	8.2	ns
2-ethyl-hexanol	40.8	2.7	68.5	31.1	64.2	15.3	63.4	5.9	ns
**Terpenes + C13 Norisoprenoids**									
a-terpineol	39.5	5.9	79.1	25.1	67.0	13.9	73.1	11.4	ns
E-8-hydroxy linalool	57.3 a	8.5	184.3 b	45.4	166.5 b	39.4	143.4 b	11.6	**
3-hydroxy-7,8-dihydro-b-ionol	57.7	27.8	85.9	49.7	51.2	0.9	51.9	13.7	ns
3-oxo-7,8-dihydro-a-ionol	62.0	15.4	83.9	21.3	100.2	40.3	74.6	24.9	ns
**Esters + Acetates**									
Isoamylacetate	116.4	15.8	200.4	38.8	279.8	45.5	178.4	15.1	ns
Ethyl hexanoate	140.5	28.8	268.3	88.2	220.8	16.0	208.5	30.0	ns
Hexyl acetate	0.0 a	0.0	0.0 a	0.0	56.8 b	22.8	53.5 b	14.2	**
Ethyl lactate	812.4	37.4	1708.7	676.4	1413.6	426.4	1059.4	49.0	ns
Ethyl octanoate	77.8	1.0	127.9	42.3	113.1	16.1	93.7	19.9	ns
Diethyl succinate	6128.4 a	373.3	13,403.3 b	3146.3	10,634.5 ab	1543.2	11,683.6 b	503.5	**
2-phenylethylacetate	32.7	3.9	62.0	20.8	56.7	9.5	59.1	3.4	ns
Diethyl malate	1701.6 a	61.9	3484.5 b	1019.1	2866.3 ab	580.8	2882.1 ab	191.0	*
Ethyl myristate	101.5 a	3.0	240.8 b	53.7	192.5 b	19.8	221.9 b	16.3	**
**Volatile acids**									
Isobutyric acid	107.5	4.4	209.8	74.1	208.5	57.0	196.5	24.4	ns
2+3-methylbutanoic acid	214.9	13.1	393.7	156.0	347.1	122.9	269.3	34.4	ns
Hexanoic acid	317.1	10.5	562.3	172.4	536.1	119.3	439.2	26.4	ns
Octanoic acid	597.3 a	30.4	1124.4 b	241.6	1041.5 b	148.3	787.5 ab	35.0	**
Nonanoic acid	687.7 a	25.0	1474.3 b	393.7	1312.9 b	213.0	1456.5 b	88.8	**
Decanoic acid	197.3 a	19.8	437.2 b	117.9	366.1 ab	59.6	307.8 ab	32.9	*
Hexadecanoic acid	79.9	22.4	151.6	71.5	127.0	49.1	102.1	2.9	ns
**Aldehydes**									
Phenylethanal	67.9	5.3	88.8	27.3	88.7	13.0	71.5	3.2	ns
**Volatile Phenols**									
4-ethylguaiacol	51.3 a	3.1	37.3	7.4	41.2	13.3	48.4	6.4	ns
4-ethylphenol	118.1 b	7.9	0.0 a	0.0	0.0 a	0.0	0.0 a	0.0	***
4-vinylguaiacol	40.6	4.2	70.1	40.6	76.3	19.1	67.7	4.2	ns
4-vinylphenol	375.1 a	28.9	910.6 b	237.6	711.2 ab	84.8	773.4 b	45.4	**
**Lactones**									
Butyrolactone	382.	14.7	654.1	227.7	619.9	145.8	667.2	47.8	ns

Different letters indicate significant differences for Tukey’s test at *p* < 0.05. Signs: *—significance at *p* < 0.05; **—significance at *p* < 0.01; ***—significance at *p* < 0.001; ns—not significant; CT—control; DN—defoliation on the north side of the canopy; DS—defoliation on the south side of the canopy; DNS—defoliation on both sides of the canopy.

**Table 6 plants-11-00591-t006:** Relative intensity (% I), relative frequency (% F) and geometric mean (% GM) determined for the sensory descriptors of Aglianico red wines obtained by early basal leaf removal treatments.

Descriptors		CT			DN			DS			DNS		
		%I	%F	%GM	%I	%F	%GM	%I	%F	%GM	%I	%F	%GM
**Odor**	Intensity	63.5	100.0	79.7	64.5	100.0	80.3	67.0	100.0	81.8	70.0	100.0	**83.7**
	Quality	54.5	85.0	68.1	63.5	100.0	79.7	61.5	100.0	78.4	71.0	100.0	**84.3**
	Fruity	50.5	90.0	67.2	52.0	90.0	**68.4**	45.0	85.0	61.7	54.5	85.0	68.1
	Floral	19.0	50.0	30.8	26.5	65.0	41.5	24.5	65.0	39.9	26.0	70.0	**42.6**
	Herbaceous	25.0	55.0	37.1	23.5	70.0	**40.5**	19.5	70.0	36.7	19.0	70.0	36.4
	Red fruts	46.5	85.0	62.8	49.0	90.0	66.4	55.0	100.0	74.1	61.0	95.0	**76.1**
	Spicy	31.5	70.0	46.9	38.5	85.0	57.2	38.0	90.0	58.4	43.5	90.0	**62.5**
	Phenolic	21.5	60.0	35.8	25.0	60.0	38.4	27.5	70.0	**43.9**	26.0	55.0	37.8
	Mint	3.5	5.0	4.2	-	-	-	2.5	5.0	3.5	-	-	-
	Balsamic	5.5	10.0	7.4	-	-	-	2.5	5.0	3.5	-	-	-
	Earthy	3.0	5.0	3.9	-	-	-	2.5	5.0	3.5	-	-	-
	Ripe fruit	4.5	5.0	4.7	-	-	-	3.0	5.0	3.9	-	-	-
	Liquor	2.0	5.0	3.2	-	-	-	-	-	-	8.5	10.0	9.2
	Peper	-	-	-	2.5	5.0	3.5	-	-	-	-	-	-
	Raisin	2.5	5.0	3.5	1.0	5.0	2.2	5.0	10.0	7.1	-	-	-
	Vanilla	-	-	-	-	-	-	-	-	-	5.5	10.0	7.4
	Licorice	-	-	-	-	-	-	1.0	5.0	2.2	4.5	10.0	6.7
**Taste**	Quality	54.5	95.0	71.9	53.0	90.0	69.1	51.5	90.0	68.0	63.0	95.0	**77.4**
	Sweetness	20.0	50.0	31.6	22.5	55.0	35.2	26.0	65.0	41.0	30.0	70.0	**45.8**
	Saltiness	17.0	60.0	31.9	18.5	80.0	38.5	22.5	80.0	**42.4**	19.5	50.0	31.2
	Acidity	46.5	90.0	**64.7**	41.5	80.0	57.6	44.0	90.0	62.9	44.0	95.0	**64.7**
	Bitterness	41.0	100.0	64.0	41.0	100.0	64.0	37.0	95.0	**59.1**	36.5	95.0	58.9
	Astringency	53.0	95.0	70.9	59.0	100.0	**76.8**	53.0	100.0	72.6	51.0	95.0	73.6
	Body	49.5	90.0	66.7	52.5	95.0	70.6	53.0	100.0	72.8	60.0	100.0	**77.5**
	Persistence	54.0	100.0	73.5	52.0	95.0	70.3	54.0	100.0	73.5	64.0	100.0	**80.0**
	Balanced	-	-	-	-	-	-	3.0	5.0	3.9	4.0	5.0	4.5
	**Total quality**	59.0	100.0	76.8	61.0	100.0	78.1	58.0	100.0	76.1	67.5	100.0	**82.1**

CT—control; DN—defoliation on the north side of the canopy; DS—defoliation on the south side of the canopy; DNS—defoliation on both sides of the canopy.

## Data Availability

Not Applicable.

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
