# Peer review of "Effect of Early Basal Leaf Removal on Phenolic and Volatile Composition and Sensory Properties of Aglianico Red Wines"

_plants, 2022, doi:10.3390/plants11050591_

Round 1

Author Response

Dear Reviewer 1

All answers to the reviewers’ comments and changes were made in blue color in the manuscript to visualize better these modifications. We hope that this new version of our manuscript would reach the high quality standards for being published as a research paper in Plants.

Looking forward to hearing from you.

Sincerely yours,

Mar Vilanova

Reviewer 1

The authors studied the influence of early basal leaf removal (15 days before flowering) on Aglianico wines quality produced in Apulia region (Italy). Instrumental (HPLC-DAD-MS and GC-MS) and sensory analysis (QDA) were used to evaluate the treatment effect on phenolic and volatile composition and sensory descriptors of wines.

The paper can be interesting but it needs major revisions.

1-Please see the instructions for authors: the Abstract is too long, there are 336 words, but it should be a total of about 200 words maximum.

The abstract was reduced as the reviewer has recommended (Now Lines 19-30)

2-Please, check the correct use of the tenses of the English verbs in all the paper.

The English was revised

3-The authors used the term “aroma” but it would be better use “odour”, in sensory analysis aroma is generally used to define the odour perception in mouth (retronasal perception).

It is true. The term “aroma” was changed by “odour” as the reviewer has recommended. However, the term aroma is usually used to Olfactory sensations (Meilgaard, Civille and Car, 1999)

4- Revise the numbers of references, they are put twice. (1. 1, 2.2, etc)

The numbers of references were revised.

5-The main comment is about the confusion about tasting and sensory analysis and hedonic evaluation.

The authors wrote that the wines were subjected to sensory analysis (QDA) but there are some mistakes.

In descriptive sensory analysis the panel characterize the products using qualitative attributes and measuring the intensities of these attributes. The first step is to individualize the attributes in all the products to be described. The panel defines the terms (attributes or descriptors), selects reference standards for each term and a score for each reference. Then the attributes are measured in all the set of samples to be described. The attributes should be non-hedonic, singular, non-redundant, and able to describe the differences among the samples. In your case the panel did not define the terms and no reference standards were identified for each attribute. They used terms with a hedonic evaluation: quality of aroma, quality of taste and total quality. These are not attributes but hedonic evaluations. “Balanced” (it should be Balance) is also a hedonic evaluation and they can’t be included in the list of the attributes. What do the authors mean with total quality? It is not possible to use this term in descriptive sensory analysis, (it is used in tasting wines, not in sensory analysis). For the taste the correct terms for the attributes should be: sweetness, saltyness, bitterness, and acidity. For example, the attribute to be quantified is sweetness, saltyness, bitterness, and acidity. The authors did not identify a reference standard specific for each term that anyone can reprepare in the same way. For example, for vanilla you could use as reference standard a solution of vanillin of known concentration.

We use the term sensory analysis as tasting in our woks because OIV define sensory analysis or evaluation as tasting (OIV. Review document on sensory analysis of wine, 2015).

The development of the sensory analysis was made in collaboration with Rias Baixas AOC and their trained panel. In this sense, the trained panel from Rias Baixas AOC generated a set of terms that describe Aglianico wines. These terms are usually used by Rias Baixas wines. evaluation.

Definition of the terms used in this study is part of the panel training made by Rias Baixas AOC with their panel during years, and the panel leader to evaluate individual judges is from Rias Baixas AOC. Also, this panel is evaluated by ENAC (Entidad Nacional de Acreditación, Spain) to develop the sensory analysis and certified the quality of wines.

On the other hand, we have included the term aroma and taste quality and total quality (global perception of the wine) because this panel always use this is a term in the evaluation of Rias Baixas AOC wines. The aroma descriptors used in this work were selected by the panel before wine tasting some samples of Aglianico wines. Then, the first part was to define the descriptors of Aglianico wines. In the second part of the analysis the list was reduced and 15 odour descriptors after eliminate redundances or synonym terms.

To clarifies this analysis, more explanation was included in materials and methods

For the taste analysis the correct terms for taste attributes (sweetness, saltiness, bitterness, and acidity) were included in the table 6 and Figure 3.

Other observations:            

6-Line 460: ...sensory properties of the wines were improved.

This is not correct, better: As a consequence, the sensory attribute intensities were influenced too.

It was done.

7-line 83-84 Instrumental and sensory analyses are tools that permit to know the phenolic and aromatic quality of the wines.

Better: Instrumental and sensory analyses allow to study the phenolic and aromatic composition of the wines..

It was done.

8-Line 84-87

Chromatographic techniques are an important analytical tool for volatile and non-volatile components of the wine, although the sensory impact of non-volatiles and volatiles identified is evaluated generally by considering perception thresholds [26].

Please modify this sentence, its meaning is not clear and it is not possible for human nose to perceive non-volatile compounds.

Maybe do you mean this: Volatile and non-volatile components of the wine can be identified and quantified by chromatographic techniques, and the sensory impact of volatiles depends on their perception thresholds [26].

It was done.

9- revise the sentence line 87-89: what do you means with “invoice”?

It was done. The sentence was changed.

10-Line 89-90 revise this sentence, Sensory descriptive analysis is not a tool, but a sensory methodology

It was done. The sentence was changed.

11-line 91: descriptions...maybe description

It was done.

12-line 91-92 Please modifiy this sentence, the instrumental and sensory data give us information on the composition of wines, but not on the quality.

Information from the two different types of tools, instrumental and sensory data, is very important to establish the quality of wine and this relationship has been explored by other researchers Better: The relationship between instrumental and sensory data were extensively studied [15,25,26,29-31]

It was done.

13- I suggest the authors to consider also this paper:

Guidoni, S., Oggero, G., Cravero, S., Rabino, M., Cravero, M. C., & Balsari, P. (2008). Manual and mechanical leaf removal in the bunch zone (Vitis Vinifera L., cv Barbera): effects on berry composition, health, yield and wine quality, in a warm temperate area. OENO One, 42(1), 49–58. https://doi.org/10.20870/oeno-one.2008.42.1.831R2

This reference was now considered in introduction and results and discussion sections.

Reviewer 2 Report

The reviewed paper (Manuscript ID plants-1588464) studied the effect of early basal leaf removal on the levels of phenolic and volatile compounds and sensory properties of Aglianico red wines. The following questions/issues need to be carefully revised/addressed before acceptance for publication.

  1. Lines 64-65, please cite relevant references to support this statement.
  2. Lines 101, two-way ANOVA? From my understanding, there is only one variable (treatment), so how did you carry out two-way ANOVA analysis?
  3. Table 3. How do you confirm the compounds are indeed correctly identified when chemical standards are not used? Moreover, the sample were analyzed using nominal MS without high mass accuracy measurements. The high-resolution mass spectrometry should be used for identification, as this instrument can distinguish between compounds with the same nominal mass. Strictly speaking, the compounds listed in Table 3 are tentatively identified. Please clarify this in the text.
  4. Figure 2 caption: please check the unit of volatile concentration, µg/L or g/L? Please show the results of statistical analysis in figure.
  5. Table 5, please confirm the identification of the compounds. Some of them are probably not correct, e.g., 2+3-methyl-1-butanol, 2+3-methylbutanoic acids, phenylettan….
  6. Lines 303-305, please rephrase the sentence.
  7. Table 6 and Figure 3, Fenolic? Does this refer to Phenolic? If so, please use phenolic instead.
  8. Lines 430-432, Please give more detailed information about panelists, i.e., age, gender, etc. If all panelists were well trained before sensory evaluation?
  9. Lines 459-461 “Defoliation led to wines with higher phenolic contents and higher concentrations of volatile compounds. As a consequence, the sensory properties of the wines were improved”. Please rephrase these two sentences. The improved sensory properties of wine samples are the phenomenon that you have observed from your sensory evaluation, not the consequence of the increased concentration of volatiles. Please note that the higher level of volatile compounds does not always beneficial for the sensory characteristics, as there are some volatile compounds with unpleasant odors, such as some volatile phenols.

Author Response

All answers to the reviewers’ comments and changes were made in blue color in the manuscript to visualize better these modifications. We hope that this new version of our manuscript would reach the high quality standards for being published as a research paper in Plants.

Looking forward to hearing from you.

Sincerely yours,

Mar Vilanova

The reviewed paper (Manuscript ID plants-1588464) studied the effect of early basal leaf removal on the levels of phenolic and volatile compounds and sensory properties of Aglianico red wines. The following questions/issues need to be carefully revised/addressed before acceptance for publication.

1.    Lines 64-65, please cite relevant references to support this statement.
It was done

2.    Lines 101, two-way ANOVA? From my understanding, there is only one variable (treatment), so how did you carry out two-way ANOVA analysis?
Sorry for this mistake. This sentence was now corrected.

3.    Table 3. How do you confirm the compounds are indeed correctly identified when chemical standards are not used? Moreover, the sample were analyzed using nominal MS without high mass accuracy measurements. The high-resolution mass spectrometry should be used for identification, as this instrument can distinguish between compounds with the same nominal mass. Strictly speaking, the compounds listed in Table 3 are tentatively identified. Please clarify this in the text.
Please consider that a column was added to Table 3, indicating the compounds whose identification was confirmed by a standard and those that were tentatively identified base don literatura data. Please see also lines 470-475 in Materials and Methods section

4.    Figure 2 caption: please check the unit of volatile concentration, µg/L or g/L? Please show the results of statistical analysis in figure.
It was done

5.    Table 5, please confirm the identification of the compounds. Some of them are probably not correct, e.g., 2+3-methyl-1-butanol, 2+3-methylbutanoic acids, phenylettan….
It was done

6.    Lines 303-305, please rephrase the sentence.
This sentence was rewritten as the reviewer has recommended (Now Lines 353-358)

7.    Table 6 and Figure 3, Fenolic? Does this refer to Phenolic? If so, please use phenolic instead.
It was done

8.    Lines 430-432, Please give more detailed information about panelists, i.e., age, gender, etc. If all panelists were well trained before sensory evaluation?
It was done

9.    Lines 459-461 “Defoliation led to wines with higher phenolic contents and higher concentrations of volatile compounds. As a consequence, the sensory properties of the wines were improved”. Please rephrase these two sentences. The improved sensory properties of wine samples are the phenomenon that you have observed from your sensory evaluation, not the consequence of the increased concentration of volatiles. Please note that the higher level of volatile compounds does not always beneficial for the sensory characteristics, as there are some volatile compounds with unpleasant odors, such as some volatile phenols.
This sentence was rewritten as the reviewer has recommended (Now Lines 538-539)

Reviewer 3 Report

Dear authors, here are some comments on the article:

Lines 223-224: „ ... increase the concentration of all volatile families of compounds with ex- ception of lactones…” . Here I consider that “and volatile phenols”  should be added. This can be seen in Figure 2.

Line 340: (a) Aroma descriptors should be changed with Taste descriptors (b) ! The last one is missing.

Line 343:   “4. Materials and Methods”  is chapter no 3 not 4. Please modify.

Line 345: In parentheses Vitis vinifera must be written in italics. The same goes for line 363 Saccharomyces cerevisiae.

Line 372: Enological parameters is subchapter 3.2 not 2.2. Please modify.

Finnaly, it is recommended to revise the format and style requested by the journal, there are different forms used in the document for tables.

Author Response

All answers to the reviewers’ comments and changes were made in blue color in the manuscript to visualize better these modifications. We hope that this new version of our manuscript would reach the high quality standards for being published as a research paper in Plants.

Looking forward to hearing from you.

Sincerely yours,

Mar Vilanova

Dear authors, here are some comments on the article:

Lines 223-224: „ ... increase the concentration of all volatile families of compounds with ex- ception of lactones…” . Here I consider that “and volatile phenols”  should be added. This can be seen in Figure 2.

Figure 2 only shows a tendency to increase lactones concentration when earlky defoliation was applied in North side (DN), not volatile phenols

Line 340: (a) Aroma descriptors should be changed with Taste descriptors (b) ! The last one is missing.

Sorry for this mistake. It was changed.

Line 343: “4. Materials and Methods”  is chapter no 3 not 4. Please modify.

It was done.

Line 345: In parentheses Vitis vinifera must be written in italics. The same goes for line 363 Saccharomyces cerevisiae.

It was done.

Line 372: Enological parameters is subchapter 3.2 not 2.2. Please modify.

It was done.

Finnaly, it is recommended to revise the format and style requested by the journal, there are different forms used in the document for tables.

It was done.

Round 2

Reviewer 1 Report

Dear Authors

the paper can be accepted now, with some minor revisions.

I am sorry, but I don't agree with the authors, sensory analysis and tasting are not the same, and the authors misunderstand the OIV document.

Abstract Line 29: The highest value for total quality was reached by DNS treatment

Do you mean global quality? (see line 349)

I suggest to remove this sentence from the abstract, 

You used terms with a hedonic evaluation: quality of aroma,  quality of taste and total quality. These are not attributes but hedonic evaluations. “Balanced” (it should be Balance) is also a hedonic evaluation and they can’t be included in the list of the attributes.

Line 317: why do you change total quality with global value? do you mean "global quality"?

“Balanced” (it should be Balance) is also a hedonic evaluation and the authors should not include it in the list of the attributes.

Balanced: with this term you mean the wine is balanced, but the correct term should be balance, because the panel evaluated the intensity of the balance in wines

Line 349 Gloabl quality ...global quality

You used terms with a hedonic evaluation: quality of aroma,  quality of taste and total quality. These are not attributes but hedonic evaluations. 

Please, explain in materials and methods the evaluation criteria used by the  panel to evaluate "odor quality, taste quality and global quality." 

It is not clear how the panel evaluated this quality. 

Author Response

Dear Reviewer

On behalf of all authors, I would like to resubmit the second revision of the manuscript entitled: Effect of early basal leaf removal on phenolic and volatile composition and sensory properties on Aglianico red wines by Debora Iorio, Giuseppe Gambacorta, Luigi Tarricone, Mar Vilanova, and Vito Michele Paradiso

The reviewer comments were made in blue color in the manuscript to visualize better these modifications. We hope that this new version of our manuscript would reach the high-quality standards for being published as a research paper in Plants.

Sincerely yours

Mar Vilanova

The paper can be accepted now, with some minor revisions.

I am sorry, but I don't agree with the authors, sensory analysis and tasting are not the same, and the authors misunderstand the OIV document.

Abstract Line 29: The highest value for total quality was reached by DNS treatment

Do you mean global quality? (see line 349)

Sorry for the mistake. It was change by total quality.

The sensory panel define total quality as the global perception of the wines (Odour and Taste)

Total quality value is according the values for Odour and Taste.

I suggest to remove this sentence from the abstract, 

We think it is important to show which one was the wine that reached the highest value in both odour and taste. However, we have deleted this sentence as the reviewer has recommended

You used terms with a hedonic evaluation: quality of aroma,quality of taste and total quality. These are not attributes but hedonic evaluations. “Balanced” (it should be Balance) is also a hedonic evaluation and they can’t be included in the list of the attributes.

This sensory analysis was made with the Rias Baixas AOC. Definition of the terms used in this study is part of the panel training made by Rias Baixas AOC with their panel during years, and the panel leader to evaluate individual judges is from Rias Baixas AOC. Also, this panel is evaluated by ENAC (Entidad Nacional de Acreditación, Spain) to develop the sensory analysis and certified the quality of wines.

On the other hand, we have included the term aroma and taste quality and total quality (global perception of the wine) because this panel always use this is a term in the evaluation of Rias Baixas AOC wines.

Line 317: why do you change total quality with global value? do you mean "global quality"?

Sorry, it is the same term.

“Balanced” (it should be Balance) is also a hedonic evaluation and the authors should not include it in the list of the attributes.

The panel of Rias Baixas AOC define Balance as Harmony, the integration of acidity, sugar, alcohol and bitter. We have included the definition in materials and methods.

Balanced: with this term you mean the wine is balanced, but the correct term should be balance, because the panel evaluated the intensity of the balance in wines

Line 349 Gloabl quality ...global quality

You used terms with a hedonic evaluation: quality of aroma, quality of taste and total quality. These are not attributes but hedonic evaluations. 

Please, explain in materials and methods the evaluation criteria used by the panel to evaluate "odor quality, taste quality and global quality." 

It is not clear how the panel evaluated this quality. 

We have added a new sentence to explain the use of these terms:

The terms balance, odour quality, taste quality and total quality, normally used by the Rias Baixas AOC panel, were added to the sensory analysis. Odour quality” and taste quality were defined by the panel as absence of defects; Total quality as the global perception of the wines (odour and taste) and balance was defined as harmony, the integration of acidity, sugar, alcohol and bitter. (Lines 512-515)

Reviewer 2 Report

The authors have put considerable effort into addressing the
reports of the referee. The paper is very much improved and I
have no problem in recommending it for publication. 

Author Response

Thank you for your revision